# Towards Identifiability of Interventional Stochastic Differential Equations

## Abstract

We study identifiability of stochastic differential equations (SDE) under multiple interventions. Our results give the first provable bounds for unique recovery of SDE parameters given samples from their stationary distributions. We give tight bounds on the number of necessary interventions for linear SDEs, and upper bounds for nonlinear SDEs in the small noise regime. We experimentally validate the recovery of true parameters in synthetic data, and motivated by our theoretical results, demonstrate the advantage of parameterizations with learnable activation functions in application to gene regulatory dynamics.

## 1 Introduction

Stochastic dynamical systems are ubiquitous as models for natural data. They are perfectly suited for application to time-series data, and therefore also a good candidate to characterize systems that reach a steady state in the limit. If a system is governed by some stochastic differential equation (SDE) and the same system is observed under different interventions, ideally one would learn the underlying parameters governing the dynamics, and guarantee accurate prediction under new interventions.

However, in many natural settings, data is modeled as following and SDE even if one does not have access to explicit trajectories. Studies of ecological systems focus on the long-term survival of multiple species modeled by the quasi-stationary state of SDEs with environmental factors as perturbations (Hening & Li, 2021). The application of flow cytometry to protein signaling networks under perturbation (Sachs et al., 2005) is destructive and yields protein quantification at one time point, modeled using the stationary distributions of linear SDEs in Varando & Hansen (2020).

One highly motivating application is single-cell genomic sequencing with high-throughput CRISPR perturbations. Biologists are often interested in inferring the gene regulatory network (GRN) that characterizes the dynamics of gene expression, informing which genes should be targeted for treatment (Dixit et al., 2016). But the destructive nature of sequencing makes it impossible to observe the trajectory of a single cell at multiple time-points, and in general it is difficult to obtain any time-series genomics data due to the high expense. Therefore, practitioners often only collect data at the end of an experiment, i.e., from the stationary distribution of the system.

Understanding the dynamics is essential for extrapolating to unseen settings, but noise and latent confounding makes it non-trivial to determine the true dynamics. Causal disentanglement aims to learn causal factors in spite of these confounders, mainly focusing on directed acyclic graph (DAG) based methods. To demonstrate these methods are well-founded, there is considerable effort devoted to understanding which models have identifiability guarantees (Lachapelle et al., 2022). However, these models suffer from inherent weakness, in particular 1) being unable to represent cycles or 2) approximate continuous-time dynamical models.

There has been renewed interest in modeling with stochastic differential equations (SDE) directly (Peters et al., 2022). In the genomic context, there is precedent for this type of modeling to represent the so- called "Waddington landscape" (Waddington, 2014), the hypothetical energy surface of cells. Furthermore, SDEs are commonly used for simulating transcriptomic datasets from a given gene regulatory network (Pratapa et al., 2020; Dibaeinia & Sinha, 2020).

As demonstrated in the context of diffusion models, SDEs are fully expressive in practice and can accurately generate observational data (Song et al., 2021). But to identify the true underlying SDE re-

quires assumptions on the model. Foundational theoretical works on identifying dynamical systems typically learn from many trajectories or even the infinitesimal generator of the dynamics (Hansen & Sokol, 2014), leaving open the harder setting of observing only the stationary distribution.

Our interest in this work is to verify which parametric assumptions are necessary for dynamical systems to have identifiability guarantees without trajectory data. Namely:

*How many interventions are necessary to identify the parameters of a stochastic differential equation, only given access to the stationary distribution?*

**Contributions**    In this work, we offer the first analysis of identifiability of interventional stochastic differential equations, with data restricted to the stationary measure. Specifically:

- We characterize tight bounds on the number of interventions necessary for identifiability of linear SDEs with shift interventions.
- We extend this analysis to nonlinear SDEs in the small noise regime, showing that identifiability is possible even without knowing the activation function of the true model.
- We apply this insight to synthetic data and semi-synthetic genomic data to confirm the efficacy of learned activations in causal SDEs, which improve expressiveness without sacrificing a simple structure and enable the inference of gene regulatory networks.

## 2 SETUP

### 2.1 NOTATION

We will write the elementary basis vectors as $\{e_i\}_i$. We will consider $\sigma : \mathbb{R}^n \to \mathbb{R}^n$ as any elementwise function, i.e. $\sigma_i(x) = \sigma_i(x_i)$. This includes elementwise activations as they are applied in multilayer perceptrons (MLPs), but we also allow for elementwise functions where each component acts differently. Writing $\sigma'$ will, unless otherwise described, denote the map $\mathbb{R}^n \to \mathbb{R}^n$ that applies elementwise the derivative of each component function, i.e. $\sigma'(x) = [\sigma'_1(x_1), \ldots, \sigma'_n(x_n)]$. We will use $Jf$ to denote the Jacobian of a vector valued function $f$, and $\Delta f$ to denote the Laplacian of $f$.

We let $P_A$ denote the orthogonal projection onto the image of matrix $A$, and $P_A^\perp = I - P_A$ the orthogonal projection onto its complement. We let $A^\dagger$ denote the pseudoinverse of $A$, and note that if $A$ has linearly independent columns it is a left inverse such that $A^\dagger A = I$, likewise for rows and right inverse. We let $\|\cdot\|$ denote the spectral norm and $\|\cdot\|_F$ the Frobenius norm. We will use $\lesssim$ to denote inequality up to constant factor, and iid as independent and identically distributed.

### 2.2 STOCHASTIC DIFFERENTIAL EQUATIONS

We consider SDEs of the following form, where $B_t$ is standard Brownian motion:

$$dX_t = v(X_t)dt + \sqrt{\epsilon}dB_t. \tag{1}$$

We will only consider autonomous systems, i.e. where the drift and noise terms have no dependence on time $t$. We will enforce the weak conditions on the drift and noise to guarantee a unique stationary distribution (Berglund, 2021), with a density $p$ that satisfies the Fokker-Planck equation,

$$0 = -\nabla \cdot (pv) + \frac{\epsilon}{2}\Delta p. \tag{2}$$

### 2.3 LINEAR SDES

We need some classical facts about linear SDEs, which are better understood than their nonlinear cousins, since Fokker-Planck can be solved explicitly.

**Theorem 2.1** (Särkkä & Solin (2019)). *Consider the SDE*

$$dX_t = (LX_t + c)dt + QdB_t \tag{3}$$

*Assume $L$ is Hurwitz, i.e., all its eigenvalues have strictly negative real parts, and $Q$ is full rank. Then the unique stationary distribution is $\mathcal{N}(-L^{-1}c, \omega)$ where $\omega$ is the unique solution to the Lyapunov equation,*

$$L\omega + \omega L^T + QQ^T = 0. \tag{4}$$

## 2.4 Interventional SDEs

We focus on the setting where we only observe the SDE through the induced stationary density under $k$ different shift interventions. Specifically, there are vectors $\{c_i\}_{i=1}^k$ with each $c_i \in \mathbb{R}^n$, and we observe the stationary distribution of the SDE,

$$dX_t = (v(X_t) + c_i)dt + \sqrt{\epsilon}dB_t. \tag{5}$$

We denote the concatenated intervention column vectors by the matrix $C \in \mathbb{R}^{n \times k}$. In the causal disentanglement literature, shift interventions typically give fewer guarantees (Buchholz et al., 2024; Squires et al., 2023). Therefore, we assume knowledge of the interventions:

**Assumption 2.2.** The shift interventions $C$ are observed.

## 3 Related Work

### 3.1 Causal Representation

In terms of modeling causality (Pearl, 2009), the most popular underlying model is the structural causal model (SCM), which characterizes the conditional distribution of a random variable under arbitrary intervention. Learning an SCM typically requires very strong assumptions such as sparsity (Schölkopf et al., 2021), interventional data (Lachapelle et al., 2022), parametric assumptions (Peters & Bühlmann, 2014), among many other results. Sparsity is a very common theme in these models, though it may also be expressed in an assumption that the number of latent variables is small, i.e. a low-rank constraint (Fang et al., 2023).

### 3.2 Causal Disentanglement with Interventional Data

The bulk of the literature on causal disentanglement focuses on SCMs with an underlying DAG. Relevant to our work are results that assume access to multiple interventional environments, either acting directly on the observed variables (Brouillard et al., 2020) or identifying a latent model under some distributional assumptions (Lachapelle et al., 2022; Squires et al., 2023; Buchholz et al., 2024).

Some of these works have addressed the crucial limitation of acyclicity (Zheng et al., 2018; Lee et al., 2019; Atanackovic et al., 2023), but without necessarily incorporating dynamics. Many works require hard interventions, where an intervened variable is a function of exogenous noise, although some can handle soft interventions (Zhang et al., 2024).

### 3.3 Dynamical System Methods

Previous work has considered modeling perturbations with dynamical systems (Peters et al., 2022). One can prove identifiability of SDE parameters from the generator (Hansen & Sokol, 2014) or trajectories (Guan et al., 2024). Other methods work in our harder setting where observed data is drawn from the stationary distribution under an SDE, and match a learned SDE under numerous interventions. These works focus on linear drift and intervention-dependent parameters (Rohbeck et al., 2024) or non-linear drift with shift interventions (Lorch et al., 2024). Notably, neither paper gives theory to confirm if these models are identifiable.

There are also methods specific to a particular scientific domain. In genomics, some methods act on pseudotime, an inferred notion of time from cell states when very few "real" timepoints are available (Wang et al., 2024; Hossain et al., 2024). Although harder to obtain due the destructive nature of RNA sequencing, genuine temporal data (even with very few timepoints) can also be modeled with the intent of extracting a GRN (Lin et al., 2025).

The closest work to ours proves an identifiability bound for linear SDEs under a strong sparsity assumption (Dettling et al., 2023). However, their result doesn't consider any interventional data, and the exact pattern of sparsity in the drift matrix is assumed to be known a priori, which is rarely the case in applied problems of interest. Another closely related work is Guan et al. (2024), which can simultaneously infer the drift and diffusion. However, this work applies only in an easier setting that assumes linearity, doesn't study interventions, and assumes multiple temporal marginals rather than just the stationary distribution.

# 4 MAIN RESULTS

We consider two main parameterizations of the drift as linear or a two-layer neural network (an MLP) to verify when the model may be uniquely identified up to appropriate invariances. Prior work has considered sparsity in the linear setting only (Dettling et al., 2023), but we focus on parameterizations that project to a low-dimensional space. In other words, although the ambient dimension $n$ may be large, we assume the hidden dimension $r$ is much smaller, at least $n > 2r$, and ideally the number of interventions to uniquely identify the parameters should scale with $r$.

## 4.1 LINEAR CASE

We start with the linear case and consider the parameterization $v(x) = (AB - D)x$ where $A \in \mathbb{R}^{n \times r}$, $B \in \mathbb{R}^{r \times n}$, $D \in \mathbb{R}^{n \times n}$. Clearly $AB$ is a redundant parameterization of a rank $r$ matrix, but we use this notation to contrast the non-linear setting in Section 4.2. To ensure the drift is Hurwitz and the SDE has a stationary distribution, we assume $\|AB\| \leq \gamma < 1$ and $D \succeq I$. Intuitively, $AB$ drive the dynamics while $D$ is a decay term to prevent unbounded dynamics. This parameterization is similar to the one used in Rohbeck et al. (2024), which also uses linear SDEs but considers hard interventions with having entirely new rows in the drift matrix, rather than shift interventions.

It is impossible to get a good deterministic guarantee if the dynamics and interventions are chosen adversarially, as seen in the following proposition:

**Proposition 4.1.** *In the linear setting, there exist choices for parameters $A, B, D$ and interventions $C$ such that the drift matrix $AB - D$ is not identifiable with less than $n - r$ interventions.*

The proof is provided in Appendix A. Intuitively, one can choose interventions that don't affect the system dynamics at all. But this situation is pathological, and in the linear case under some weak distributional assumptions we can show identifiability.

**Assumption 4.2.** The matrices $A$ and $B$ are almost surely (a.s.) full-rank, have spectral norm less than one, and are invariant to applying a rotation matrix on the left or the right. Each column of $C$ is drawn iid from some distribution with a density on $\mathbb{R}^n$.

We take care to explain why these assumptions are not particularly restrictive. The low-rank constraint is already enforced by the drift function so $A$ and $B$ are almost surely full-rank when drawn from any density. The spectral norm bound is necessary to guarantee the SDE has a stationary distribution. And the rotational invariance assumption encodes an uninformative prior on which low-rank subspace governs the dynamics. Two simple choices that satisfy these assumptions are sampling $A$ and $B^T$ from either the uniform measure on the Stiefel manifold (rectangular matrices with orthonormal columns) or with iid Gaussian entries, and then scaling down to ensure a spectral norm strictly smaller than one, with $C$ sampled from any density.

Additionally, for theoretical tractability, we presume knowledge of the decay term $D$. This is a stronger assumption, but plausible in some applied contexts. For example, in single-cell genomics the decay rate of genes can be estimated with external experiments (e.g., from BRIC-seq (Imamachi et al., 2014)).

**Assumption 4.3.** The decay matrix $D$ is observed.

Altogether, we can now present a nearly tight identifiability result in the linear case.

**Theorem 4.4.** *Consider the linear drift in Equation 5. Suppose $A, B$ are drawn from any density such that $\|A\|, \|B\| \leq \gamma < 1$. Then the drift $AB - D$ is identifiable a.s. with $r$ interventions, and unidentifiable a.s. with at most $r - 2$ interventions.*

See Appendix A for the proof. Naively, one might count $2nr$ unknown parameters in the entries of $A$ and $B$, and assume the $n^2$ entries of the stationary covariance $\omega$ would be enough to identify them, but this isn't the case. One needs exactly enough interventions to account for the hidden rank.

**Identifying the decay.** If diagonal decay term $D$ is not inferred in advance, learning it simultaneously is comparable to the setting of robust PCA (Candès et al., 2011) or recovery of a diagonal plus a positive semidefinite low rank term (Saunderson et al., 2012). However, unlike the usual matrix completion setting where entries of the matrix are revealed uniformly at random, we have access to



Figure 1: Contour plot of the stationary SDE under different activations and interventions. Activation contractivity enforces one mode, but the linear distribution is Gaussian with fixed covariance across interventions, while the nonlinear distribution can be more expressive.

correlated low-rank measurements of the form $e_i c_j^T$ for $\{e_i\}_{i=1}^n$ the elementary basis. This setting is well studied with sub-Gaussian measurement vectors (Zhong et al., 2015), but the tools do not readily apply to our setting with non-random measurements and the constraints of the Lyapunov equation. Nevertheless, we observe in Section 5 that identifiability with an unknown diagonal decay matrix $D$ is empirically achieved, while still subject to the lower bound established in Theorem 4.4.

### 4.2 NONLINEAR CASE

The nonlinear case is more challenging, because there is no longer a nearly closed form characterization of the stationary distribution. To apply tools from above, we consider when a linearized SDE can approximately capture the true dynamics, by restricting to contractive drift and small noise.

We consider the vector field $v(x) = A\sigma(Bx) - x$, with the constraints that $\|A\|, \|B\| \leq 1$. Notably, we will not assume prior knowledge of the elementwise map $\sigma : \mathbb{R}^n \to \mathbb{R}^n$, only some assumptions:

**Assumption 4.5.** The map $\sigma \in \mathcal{C}^2$ acts on each element independently and satisfies:

1. $\max_{x \in \mathbb{R}} \sigma_i'(x) = \gamma < 1$

2. $\min_{x \in \mathbb{R}} \sigma_i'(x) = \tau > 0$

3. $\max_{x \in \mathbb{R}} |\sigma_i''(x)| = M < \infty$

4. The set $\{x \in \mathbb{R} : \sigma_i''(x) = 0\}$ is measure zero.

The strongest constraint here is condition 1, upper bounding the first derivative of the activation, as it implies the noiseless dynamics are globally contractive, but it guarantees a stationary distribution exists for any intervention vector $c$. Furthermore, these conditions still allow for a wide class of possible activations $\sigma$, mainly constrained to be increasing with bounded first and second derivative. We observe the improved expressiveness of the stationary distribution of nonlinear SDEs in Figure 1. Note constraint 4 rules out functions that are linear (or locally linear). This enables a stronger possible identifiability guarantee than only recovering $A$ and $B$ up to rotation as in the linear case.

**Noiseless Setting.** One might be tempted to simply consider the noiseless case, where the SDE reduces to an ordinary differential equation (ODE), and rather than recovering the stationary distribution one recovers instead the unique global stable point. However, the noiseless case cannot take advantage of the low-rank structure, and requires $\Omega(n)$ interventions a.s. for identifiability.

**Proposition 4.6.** *Setting $\epsilon = 0$ in equation 5, the parameters are a.s. not identifiable with fewer than $n - r$ interventions.*

The proof is in Appendix A. Intuitively, the equilibrium points of the noiseless ODE approximate the mean of the SDE for small noise (note this is not true for larger noise, for example see Ma et al.

(2015)). This suggests that in order to make use of the low-rank assumption on the dynamics, one must at least take advantage of second order moments of the SDE even in the small noise limit.

**Moments in the zero-noise limit.** As the noise converges to zero, the stationary distribution converges to a dirac centered on the global stable point, and there are no higher order moments. However, the rescaled second-order moments have a non-trivial limit as noise goes to zero:

**Theorem 4.7.** *Let $x^*$ be unique solution $v(x^*) + c = 0$, and define $L = Jv(x^*)$. If $\omega$ solves the Lyapunov equation $L\omega + \omega L^T + I = 0$, and $m_\epsilon$ and $\Sigma_\epsilon$ are the mean and covariance of the stationary distribution of Equation* (5) *intervened by c, we have for sufficiently small $\epsilon$,*

$$\|m_\epsilon - x^*\| \leq \left(\frac{\epsilon n}{1 - \gamma}\right)^{1/2}, \tag{6}$$

$$\|\Sigma_\epsilon/\epsilon - \omega\| \lesssim \frac{\epsilon^{1/2} r^{5/2} n^{1/2} M}{(1 - \gamma)^3}. \tag{7}$$

The proof is given in Appendix A. This is one instance of perturbation theory for SDEs (Gardiner, 2021; Sanz-Alonso & Stuart, 2016). Equipped with this fact, one can consider access to the stationary distribution of an SDE, specifically the first and second-order moments, and inspect what happens now as the noise goes to zero. In this case, we obtain a non-trivial identifiability bound:

**Theorem 4.8.** *Suppose we observe the first and second moments $m_\epsilon$ and $\Sigma_\epsilon/\epsilon$ of the stationary distribution of the SDE in Equation* (5)*, in the limit as $\epsilon \to 0$. Then a.s. with $r + 1$ interventions, $A$ and $B^T$ are both identifiable up to (simultaneous) permutation and scaling of their columns.*

We give a brief sketch of the proof, to capture the novel elements and how the identifiability of $A$ and $B$ appears here but not in the linear case. The zero-noise limit approximately linearizes each intervened SDE. Unlike the exactly linear case, the stationary covariance is not identical across interventions. By Theorem 4.7, the covariance under the $i$th intervention approximately matches the SDE with drift $A(J\sigma(Bx_i^*))B$ where $x_i^*$ is the unique zero of the intervened drift $v(x) + c_i$.

Sufficient variability in these fixed points (guaranteed by our assumptions on $\sigma$) enables the identification of $A$ and $B$. Suppose one had access to the drift matrices: because the Jacobian term in the drift is diagonal, a linear combination of drifts across $r$ interventions can yield a rank-one matrix, which must correspond to the outer product of a column of $A$ and a row of $B$. But we don't have access to the drift matrices directly, only the covariances they induce, and so the proof relies on finding a different set of matrices where rank-one matrices that characterize $A$ and $B$ can be isolated.

Crucially, the proof doesn't require knowledge of the activation function itself. This enables identification of the bias term absorbed by $\sigma$, but furthermore one can identify the parameters $A$ and $B$ even if each elementwise activation in $\sigma$ is distinct. This observation motivates learnable activations, which we experiment with in Section 5.

We hypothesize the constraint that the SDE drift have a single absorbing point is unnecessary. If the ODE given by $\frac{dx}{dt} = v(x) + c_i$ had multiple stable equilibria, then in the limit of small noise, the SDE stationary distribution should approach a Gaussian mixture centered around each equilibrium point. The proof only requires certain linear independence conditions of vectors induced by each fixed point, so one intervention with multiple equilibria would offer essentially the same information as additional unimodal interventions. This is consistent with the observations in Lorch et al. (2024) that nonlinear SDE models generalize well with a limited number of interventions.

## 5 EXPERIMENTS

To validate the theory given above, we consider experiments demonstrating the recovery of SDE parameters and generalization to unseen interventions. To show the broad applicability of the theory, we consider multiple loss functions: a loss acting directly on the parameters in the linear case (Rohbeck et al., 2024), a kernelized Stein discrepancy (Barp et al., 2019), and rollout loss for neural SDEs (Kidger et al., 2021).

## 5.1 LINEAR SDE RECOVERY

Because we are fitting linear SDEs where the stationary distribution is Gaussian, we can choose a very simple loss that matches the population mean and covariance of each intervented distribution, to verify the claim of identifiability (Theorem 4.4). This loss is akin to the one used in the Bicycle method introduced in Rohbeck et al. (2024) that fits interventional linear SDEs. Assuming the noise scale $\epsilon$ is known and $L$ denotes the true drift, we fit the parameters $\hat{A}$, $\hat{B}$, and $\hat{D}$. Letting our estimate for the drift be denoted by $\hat{L} := \hat{A}\hat{B} - \hat{D}$, the loss function matches the mean and covariance of each intervention:

$$\mathcal{L}_{lin}(\hat{A}, \hat{B}, \hat{D}) = \|\hat{L}^{-1}C - L^{-1}C\|_F + \|\hat{L}\omega + \omega\hat{L}^T + \epsilon I\|_F. \tag{8}$$

We evaluate the recovery of $AB$ under different numbers of interventions for two different ambient dimensions $n$ and two different true ranks $r$ in Figure 2. For each sampled SDE, we use the best train error from 100 independent initializations to deal with the nonconvexity of the objective. The only exception is the oversampled $k = r\log(n)$ where we need only 5 initializations, as the landscape is empirically easier to learn. Further details are given in Section B.

We confirm our theory: when the decay is known, $r - 2$ interventions fails to grant identifiability with poor performance and extremely high variance, even with many independent initializations, while $r$ interventions clearly succeed. When the decay isn't known, we use a modest oversampling of $r\log(n)$ interventions. This amount is informed by the literature on matrix completion, where $\Omega(rn\log(n))$ revealed matrix entries were proven to be necessary for a rank $r$ matrix recovery problem (Candès & Tao, 2010). We scale down by a factor of $n$ because in our setting each intervention yields the mean vector of the perturbed linear SDE in $\mathbb{R}^n$, hence $n$ measurements.

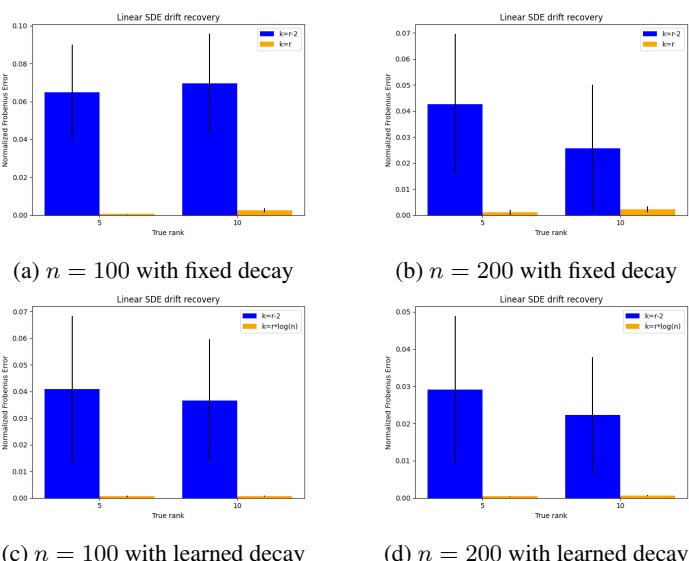

(a) $n = 100$ with fixed decay      (b) $n = 200$ with fixed decay

(c) $n = 100$ with learned decay      (d) $n = 200$ with learned decay

Figure 2: Normalized Frobenius error of learned drift against true drift in linear SDEs with $k$ independent Gaussian interventions. Error bars are standard deviation over 5 independent runs.

## 5.2 NONLINEAR SDE RECOVERY

We repeat this verification of identifiability in the setting of Theorem 4.8, using a very small value of $\epsilon = 10^{-5}$ to approximately linearize. For some randomly chosen $A$ and $B$, we define the true drift $v(x) = A\sigma(Bx) - x$ and use each intervened drift $v + c_i$ to calculate the corresponding sample mean $m_\epsilon^i$ and sample covariance $\Sigma_\epsilon^i$.

We define a parameterized drift of the form $\hat{v}_\theta(x) = \hat{A}\sigma_\theta(\hat{B}x) - x$ where $\sigma_\theta$ is a learned MLP that acts elementwise. We train with the loss function

$$\mathcal{L}_{nonlin}(\hat{A}, \hat{B}, \theta) = \sqrt{\sum_{i=1}^{k} \|\hat{v}_\theta(m_\epsilon^i) + c_i\|^2} + \sqrt{\sum_{i=1}^{k} \|J\hat{v}_\theta(m_\epsilon^i)(\Sigma_\epsilon^i/\epsilon) + (\Sigma_\epsilon^i/\epsilon)J\hat{v}_\theta(m_\epsilon^i)^T + I\|^2}$$

(9)

The motivation for this loss comes from Theorem 4.7. The first term enforces the approximate fixed point property of the mean, and the second term enforces the approximate Lyapunov equation of the covariance. Thus we can train $\hat{v}_\theta$ to match the first and second moments of the true stationary distribution of $v$, which will be approximately Gaussian for sufficiently small $\epsilon$.

To empirically evaluate identifiability, we calculate the normalized error in recovering $A$ and $B$ from the parameters $\hat{A}$ and $\hat{B}$, chosen from the run with minimum training loss across 100 independent runs. Further details are given in the Appendix. We observe that $r + 1$ interventions enable consistently good recovery of the underlying drift parameters, whereas $r$ interventions leads to high variance estimates. This suggests that the $r + 1$ upper bound proven in Theorem 4.8 may be tight, and offers a direction for further work refining the theory of identifying the nonlinear case.

Table 1: Normalized Frobenius error of drift parameters with $k$ independent Gaussian interventions, over 20 independent runs.

|  | $A$ Error | $B$ Error |
| --- | --- | --- |
| $k = r$ | $0.13 \pm 0.13$ | $0.14 \pm 0.15$ |
| $k = r + 1$ | **$0.065 \pm 0.04$** | **$0.07 \pm 0.04$** |

## 5.3 SYNTHETIC NONLINEAR SDE GENERALIZATION

For an evaluation of Theorem 4.8, we apply the insight that identifiability doesn't require knowing $\sigma$ and consider learnable MLPs. Learnable activations (Goyal et al., 2019) have been proposed before (Apicella et al., 2019) but to our knowledge not previously applied to SDE parameterizations.

To assess generalizability, we consider a loss function proposed by Lorch et al. (2024): the kernel deviation from stationarity (KDS), or equivalently the kernelized Stein discrepancy of the stationary distribution (Barp et al., 2019). When the parametric drift is defined as $v_\theta(x) = A\sigma(Bx) - Dx$ and the target stationary distribution is $\mu$, the KDS is,

$$\mathcal{L}_{KDS}(\theta) = E_{x \sim \mu} E_{x' \sim \mu} \mathcal{A}_x \mathcal{A}_{x'} k(x, x').$$

(10)

where $k$ is a given kernel and $\mathcal{A}_x$ is the SDE generator acting on the variable $x$ (for more details see Lorch et al. (2024)). We use samples from the true stationary distribution $\mu$ to approximate these expectations, further training details are given in the Appendix. This loss is nevertheless unstable when using noise small enough to reach the limit proven in Theorem 4.8, and therefore we assess generalizability according to performance on unseen interventions.

We observed numerical instability in the Sinkhorn divergence without large entropic regularization (Cuturi, 2013). Therefore, we follow Zhang et al. (2023); Lorch et al. (2024) and evaluate with the mean squared error (MSE) between the true and predicted distribution. Further details are given in Section B.3. The results are given in Table 2. We observe a clear benefit for low noise SDEs, which gets smaller as the noise gets bigger and hence further away from our proven settings. This is consistent with our theory, and also intuitive: as the noise gets larger, the stationary distribution gets smoother and has less dependence on the intervention vectors. Nevertheless, in the low noise regime we see that learnable activations improve prediction on test interventions without overfitting.

## 5.4 SIMULATED GRN SDE GENERALIZATION

We consider an applied setting for learnable activations, and how they impact the recovery of GRNs. In the genomics context, $n$ is the number of expressed genes in the thousands, and $r$ corresponds to a much smaller number of latent gene modules/pathways. Because there are few fully known regulatory networks we rely on semi-synthetic data to produce data with similar characteristics to true perturbed transcriptomic samples and a ground-truth GRN.

Table 2: Mean distance between true and predicted distribution with $n = 20$, $r = 3$, over 20 independent seeds with 20 test interventions per seed, with sigmoid activation versus learned activation.

|  | Sigmoid | Activation MLP |
|---|---|---|
| $\epsilon = 0.05$ | $21.78 \pm 12.78$ | $\mathbf{9.51 \pm 1.28}$ |
| $\epsilon = 0.1$ | $16.54 \pm 6.35$ | $\mathbf{9.23 \pm 1.65}$ |
| $\epsilon = 0.2$ | $11.53 \pm 3.47$ | $\mathbf{7.96 \pm 1.44}$ |
| $\epsilon = 0.3$ | $7.97 \pm 2.19$ | $\mathbf{6.69 \pm 2.07}$ |

We use the PerturbODE model from Lin et al. (2025), similar to our setup and modified to handle a loss for parameterized SDEs (Kidger et al., 2021). We consider the inferred GRN as the matrix multiplication of the weight matrices in the MLP that parameterizes the drift, corresponding to a first-order Taylor approximation of the dynamics. GRN recovery can be measured as a classification task of individual edges. As a baseline, we also consider the Bicycle interventional linear SDE method (Rohbeck et al., 2024). To simulate a known GRN, we use the boolean ODE based-simulator BEELINE (Pratapa et al., 2020). BEELINE uses the provided GRN to parametrize an SDE in the space of mRNA expression and protein expression, with physically plausible regulation dynamics. More precise details are given in the Appendix in Section B.4.

GRN recovery is still extremely difficult (random guessing is $\approx 0.0586$ and maximum AUPRC for dynamical methods $\approx 0.07$), but we observe improvement using nonlinear SDEs over linear SDEs, and substantial improvement using learnable activation in the SDE parameterization (Figure 3).

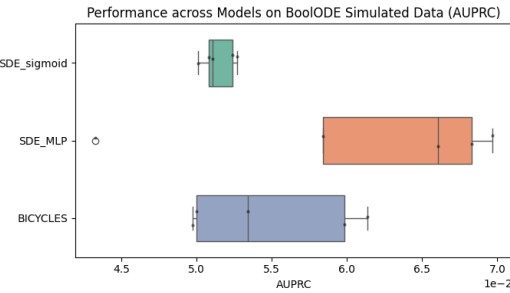

Figure 3: Gene regulatory network recovery on three tested SDE models for 5 independent runs.

## 6 DISCUSSION

We confirm identifiability directly in the linear case and non-linear, low-noise limit, and see improved generalization using learned activations for small noise. It may be beneficial to increase the capacity of the SDE beyond two-layer neural networks, if the extracted drift were still identifiable.

We note some limitations of the given theory. The results demonstrate identifiability in a setup with infinitesimal small noise and restrictions on the possible drift functions. Generalizing to larger noise regimes may be possible with higher order moment information about the stationary distribution, although understanding the stationary distribution outside of linearization is very challenging. Broader parameterizations of the drift or interventions are interesting extensions for future work.

## 7 CONCLUSION

In this work, we've given the first provable results regarding identifiability of interventional SDEs, in both the linear and nonlinear case. Although such models are currently less popular and less studied then comparable SCM-based modeling, the ability to obtain provable guarantees for dynamical systems without any temporal data suggests they may become more fruitful for causal inference in the future. As longitudinal genomics datasets often have a small number of time-points, future work may consider how to incorporate sparse trajectory information with the stationary distribution for better identifiability guarantees or more effective architectures for recovering regulatory networks.

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

## A  PROOFS OF RESULTS

PROOF OF PROPOSITION 4.1

Consider block matrices such that,

$$A = \left[\begin{array}{c|c} \frac{1}{\sqrt{2}}I_r & 0 \\ \hline 0 & 0 \end{array}\right] \tag{11}$$

$$B = \left[\begin{array}{c|c} \frac{1}{\sqrt{2}}I_r & 0 \\ \hline 0 & 0 \end{array}\right] \tag{12}$$

and set $D = I$ and $\epsilon = 1$. If we also set

$$\omega = \left[\begin{array}{c|c} I_r & 0 \\ \hline 0 & \frac{1}{2}I_{n-r} \end{array}\right] \tag{13}$$

then it is straightforward to confirm that the drift matrix $L = AB - I$ satisfies the Lyapunov equation

$$L\omega + \omega L^T + I = 0 \tag{14}$$

Now, consider any skew-symmetric matrix $Q$ that is only supported in the top left $r \times r$ block. Then $AB + Q\omega^{-1}$ will only be supported in the top left block and therefore still have rank $r$. Furthermore, $Q$ obeys the trivial Lyapunov equation $Q\omega + \omega Q^T = 0$, so we have that the drift matrix $\hat{L} := AB + Q\omega^{-1} - I$ also satisfies the Lyapunov equation $\hat{L}\omega + \omega\hat{L}^T + I = 0$. Choosing the norm of $Q$ small enough guarantees that $\hat{L}$ is still Hurwitz, while choosing $Q \neq 0$ guarantees $L \neq \hat{L}$.

In the worst case, every intervention is of the form $e_i$ for $i > r$. The block structure and Woodbury matrix identity imply that $(AB + Q\omega^{-1} - I)^{-1}e_i = -e_i$ for any $Q$ chosen as above, and therefore $-\hat{L}^{-1}e_i = -L^{-1}e_i$.

We conclude that for each intervention $e_i$ with $i > r$, by Theorem 2.1 the drift matrices $L$ and $\hat{L}$ both induce identical stationary distributions. Thus, even with $n - r$ interventions, the drift is not identifiable.

PROOF OF THEOREM 4.4

We show separately the upper and lower bounds for number of interventions needed for almost sure identifiability.

**Theorem A.1.** *Suppose the true $A$ and $B$ are sampled according to Assumption 4.2, with the constraint that a.s. $\|A\|, \|B\| \leq \sqrt{\gamma}$. Then $L = AB - D$ is a.s. identifiable.*

*Proof.* Recall that $C = [c_1, \ldots, c_r]$ is the matrix of interventions as columns. The means of the interventional stationary distributions are given by,

$$-L^{-1}C = (D^{-1} + D^{-1}A(I - BD^{-1}A)^{-1}BD^{-1})C \tag{15}$$

So with knowledge of $D$ and $C$, we can calculate $A(I - BD^{-1}A)^{-1}BD^{-1}C$. This matrix is a.s. full rank, and its range is equal to the range of $A$. Hence we can infer $P := P_A^\perp$.

Now, using the Lyapunov equation, we have,

$$0 = L\omega P + \omega L^T P + \epsilon P \tag{16}$$
$$= L\omega P - \omega DP + \epsilon P \tag{17}$$

Rearranging,

$$2\omega P = L^{-1}(\omega D + \epsilon I)P \tag{18}$$

$(\omega D + \epsilon I)P$ is rank $n - r$, so a.s. $\mathrm{im}(\omega D + \epsilon I)P \oplus \mathrm{im}\,C$ is $n$ dimensional. Hence we recover $L^{-1}$ and therefore $L$.

$\square$

**Theorem A.2.** *Sample $A \in \mathbb{R}^{n \times r}$ and $B \in \mathbb{R}^{r \times n}$ according to Assumption 4.2. Then a.s. $AB$ is not identifiable with $r - 2$ interventions.*

*Proof.* Assume the decay is fixed at $D = I$, so $L = AB - I$ with induced covariance $\omega$. With $r - 2$ interventions, $ABL^{-1}C$ is at most rank $r - 2$, and by assumption the kernel of $A^T$ is at most dimension $n - r$, so there is guaranteed to be a two dimensional subspace orthogonal to the previous spaces, say with basis vectors $u$ and $v$. Let $Q^* = uv^T - vu^T$.

Then we consider $\hat{L} = L + \omega Q$ where $Q = B^T A^T Q^* AB$, and claim that $\hat{L}$ is a valid, distinct drift matrix that generates the same data.

First, note that $u, v$ are orthogonal to the kernel of $A^T$ and $B^T$ is full-rank a.s., so $B^T A^T Q^*$ is non-zero. Transposing and applying the same reasoning, we get that $Q = B^T A^T Q^* AB \neq 0$. Scaling down the magnitude of $Q^*$ if necessary, we have that $\hat{L}$ is Hurwitz and distinct from $L$.

Second, note that $\hat{L} = (I + \omega B^T A^T Q^*)AB - I$, so it satisfies the rank $r$ constraint on the non-diagonal part.

Now we show it agrees on the induced stationary distributions. Again from the choice of the two dimensional subspace, we have that $Q^* ABL^{-1}C = 0$, which implies $QL^{-1}C = 0$.

From the Woodbury matrix identity, we see the means of all stationary distributions are conserved,

$$\hat{L}^{-1}C = (L + \omega Q)^{-1}C \tag{19}$$

$$= \left[ L^{-1} - L^{-1}\omega(I + QL^{-1}\omega)QL^{-1} \right] C \tag{20}$$

$$= L^{-1}C \tag{21}$$

Finally, note that by antisymmetry of $Q$,

$$\hat{L}\omega + \omega\hat{L}^T = L\omega + \omega L^T + \omega Q\omega + \omega Q^T\omega \tag{22}$$

$$= L\omega + \omega L^T \tag{23}$$

which implies $\hat{L}$ satisfies the same Lyapunov equation and therefore induces the same covariance $\omega$. $\qquad\square$

PROOF OF PROPOSITION 4.6

In the zero-noise setting, the SDE reduces to an ODE, and in the limit as time goes to infinity, the stationary distribution is simply a point mass on the unique fixed point of the drift. Therefore, one only gets access to the stationary points $x_i^*$ for each vector field $v + c_i$: other higher moments are not defined.

Suppose we have $n - r - 1$ interventions. Since $A$ has $r$ columns, there must be a vector $u$ in $\mathbb{R}^n$ orthogonal to the columns of $A$ and $C$.

Let $\hat{B} = B + bu^T$ for any non-trivial $b \in \mathbb{R}^r$. We claim $\hat{B}$ induces the same data as $B$. First, note that because $u$ is orthogonal to every $c_i$, and $x_i^* - c_i$ is in the image of $A$, it follows $u$ is also orthogonal to each $x_i^*$.

Thus, one can confirm that $\hat{B}x_i^* = Bx_i^*$, so the fixed points don't change. But $\hat{B} \neq B$, and they are clearly not equivalent up to permutation or scaling invariance.

PROOF OF THEOREM 4.7

We will require two lemmas.

**Lemma A.3.** *Consider a vector field $v(x) = A\sigma(Bx) - x + c$ with $\|A\|, \|B\| \leq 1$, $\|\sigma\|_\infty = \gamma < 1$. Let $p$ be the stationary distribution of our usual SDE, $x^*$ be the unique stationary point of $v$, and $P$ be an orthogonal projection such that $BP = B$ and $PA = A$. Assume $2(j - 1) \leq Tr(P)$, then,*

$$E_p \left[ \|P(x - x^*)\|^{2j} \right] \leq \left( \frac{\epsilon Tr(P)}{1 - \gamma} \right)^j. \tag{24}$$

*Proof.* Let $f(x) = A\sigma(Bx) + c$. Note $f$ is a contraction and $f(x) = f(Px)$. We have by Cauchy-Schwarz,

$$P(x - x^*) \cdot v(x) = P(x - x^*) \cdot (v(x) - v(x^*)) \tag{25}$$

$$= P(x - x^*) \cdot (f(x) - f(x^*) - (x - x^*)) \tag{26}$$

$$= P(x - x^*) \cdot (f(Px) - f(Px^*) - (x - x^*)) \tag{27}$$

$$\leq -(1 - \gamma)\|P(x - x^*)\|^2 \tag{28}$$

Choose $g_j(x) = \|P(x - x^*)\|^{2j}$, then if $T := Tr(P)$ we have,

$$\nabla g_j(x) = 2jg_{j-1}(x)P(x - x^*) \tag{29}$$

$$\Delta g_j(x) = 2jTg_{j-1}(x) + 4j(j - 1)g_{j-1}(x) \tag{30}$$

The generator of the SDE is $\mathcal{A}g = \nabla g \cdot v + \frac{\epsilon}{2}\Delta g$. So Fokker-Planck gives,

$$0 = E_p\left[\mathcal{A}g_j\right] \tag{31}$$

$$= E_p\left[\nabla g_j \cdot v(x) + \frac{\epsilon}{2}\Delta g_j\right] \tag{32}$$

$$= E_p\left[2jg_{j-1}P(x - x^*) \cdot v(x) + \frac{\epsilon}{2}(2jT + 4j(j-1))g_{j-1}(x)\right] \tag{33}$$

$$\leq E_p\left[-2j(1-\gamma)g_j + \frac{\epsilon}{2}(2jT + 4j(j-1))g_{j-1}(x)\right] \tag{34}$$

$$\tag{35}$$

Simple algebra gives,

$$E_p\left[g_j\right] \leq \frac{\epsilon(T + 2(j-1))}{2(1-\gamma)}E_p\left[g_{j-1}(x)\right] \tag{36}$$

Now apply the assumption $2(j-1) \leq T$ and the result follows by induction. $\qquad\square$

**Lemma A.4.** *Assume same conditions as previous lemma. Then the Taylor expansion with remainder around $x^*$ given by*

$$v(x) = Jv(x^*)(x - x^*) + R(x), \tag{37}$$

*satisfies the bound*

$$\|R(x)\| \leq 2\sqrt{2}r^{3/2}M\|P(x - x^*)\|^2, \tag{38}$$

*where $M := \sup_i \|\sigma_i''\|_\infty$.*

*Proof.* Assume that $\operatorname{im} A \oplus (\ker B)^\perp$ is only supported on the first $2r$ coordinates. The $i$th coordinate of the Taylor remainder must take the form,

$$R_i(x) = v_i(x) - e_i^T Jv(x^*)(x - x^*) \tag{39}$$

$$= \sum_{|\alpha|=2} \frac{2}{\alpha!}(x - x^*)^\alpha \int_0^1 (1-t)\partial_\alpha v_i(x^* + t(x - x^*))dt \tag{40}$$

By assumption, second order derivatives of $v$ are always bounded by $M$, and they are identically zero if $i > 2r$. This implies $R_i$ is only nonzero for $i \leq 2r$, in which case,

$$|R_i(x)| \leq M \sum_{|\alpha|=2, \alpha_{>2r}=0} \frac{2}{\alpha!}|(x - x^*)^\alpha| \tag{41}$$

$$\leq M \left(\sum_{j=1}^{2r} |x_j - x_j^*|\right)^2 \tag{42}$$

$$\leq 2rM \left(\sum_{j=1}^{2r} |x_j - x_j^*|^2\right) \tag{43}$$

$$= 2rM\|P(x - x^*)\|^2 \tag{44}$$

where $P$ the projection onto the first $2r$ coordinates.

Hence,

$$\|R(x)\|^2 \leq 8r^3 M^2 \|P(x - x^*)\|^4. \tag{45}$$

Now, we drop the assumption that $\operatorname{im} A \oplus (\ker B)^\perp$ is restricted to the first $2r$ elements and extend the result more generally.

Consider the orthogonal matrix $Q$ such that $\operatorname{im} QA \oplus (\ker BQ^T)^\perp$ is restricted to the first $2r$ components, then by design the above result can be applied to the new drift function

$$\tilde{v}(y) := QA\sigma(BQ^T y) - y + Qc \tag{46}$$

whose unique stationary point is $y^* = Qx^*$.

Hence, applying the above result to the remainder term $\tilde{R}(y) := \tilde{v}(y) - J\tilde{v}(y^*)(y - y^*)$, with $\tilde{P}$ the projection onto the first $2r$ components, we get

$$\|\tilde{R}(y)\| \leq 2\sqrt{2}r^{3/2}M\|\tilde{P}(y - y^*)\|^2 \tag{47}$$

If we let $y = Qx$, then some algebra reveals

$$\tilde{R}(y) = \tilde{v}(y) - J\tilde{v}(y^*)(y - y^*) \tag{48}$$

$$= QA\sigma(BQ^T y) - y + Qc - (QAJ\sigma(BQ^T y^*)BQ^T - I)(y - y^*) \tag{49}$$

$$= Q(A\sigma(Bx) - x + c - (AJ\sigma(Bx^*)B - I)(x - x^*)) \tag{50}$$

$$= Q(v(x) - Jv(x^*)(x - x^*)) \tag{51}$$

$$= QR(x) \tag{52}$$

And so we can finally conclude

$$\|R(x)\| = \|\tilde{R}(y)\| \tag{53}$$

$$\leq 2\sqrt{2}r^{3/2}M\|\tilde{P}(y - y^*)\|^2 \tag{54}$$

$$= 2\sqrt{2}r^{3/2}M\|Q^T\tilde{P}Q(x - x^*)\|^2 \tag{55}$$

Note that by our choice of $Q$, $Q^T\tilde{P}Q$ is the orthogonal projection onto $\operatorname{im} A \oplus (\ker B)^\perp$, and so we have the bound.

$\square$

Using these results, we can proceed to the main perturbation theory result:

*Proof of Theorem 4.7.* We will drop the $\epsilon$ subscript, where it is clear that $m := m_\epsilon$ and $\Sigma := \Sigma_\epsilon$ are the first and second order moments of the stationary distribution.

The mean bound follows from Lemma A.3 and Jensen's inequality,

$$\|m - x^*\| = \|E_p[x - x^*]\| \tag{56}$$

$$\leq E_p\left[\|x - x^*\|^2\right]^{1/2} \tag{57}$$

$$\leq \left(\frac{\epsilon n}{1 - \gamma}\right)^{1/2}. \tag{58}$$

Fokker-Planck yields a formula for second-order moments of the SDE stationary distribution (see for example Chapter 5.5 in Särkkä & Solin (2019)), such that,

$$0 = E_p[(x - m)v(x)^T + v(x)(x - m)^T + \epsilon I] \tag{59}$$

Note the simple fact that

$$E_p[(x - m)(x - x^*)^T] = E_p[(x - m)(x - m + m - x^*)^T] \tag{60}$$

$$= E_p[(x - m)(x - m)^T] = \Sigma \tag{61}$$

combined with the linearization of $v$ we can write,

$$0 = E_p[(x - m)(x - x^*)^T L^T + L(x - x^*)(x - m)^T + \epsilon I + (x - m)R(x)^T + R(x)(x - m)^T] \tag{62}$$

$$= L\Sigma + \Sigma L^T + \epsilon I + E_p[(x - m)R(x)^T + R(x)(x - m)^T] \tag{63}$$

and dividing by $\epsilon$,

$$0 = L\frac{\Sigma}{\epsilon} + \frac{\Sigma}{\epsilon}L^T + I + \frac{1}{\epsilon}E_p[(x-m)R(x)^T + R(x)(x-m)^T] \tag{64}$$

As in the lemmas, we let $P$ be the orthogonal projection onto $\operatorname{im} A \oplus (\ker B)^\perp$. Then by Lemma A.3, Lemma A.4 and Cauchy-Schwarz we have,

$$E_p\left[\|x-m\| \cdot \|R(x)\|\right] \lesssim r^{3/2} M E_p\left[(\|x-m\| \cdot \|P(x-x^*)\|^2\right] \tag{65}$$

$$\lesssim r^{3/2} M E_p\left[\|x-m\|^2\right]^{1/2} E_p\left[\|P(x-x^*)\|^4\right]^{1/2} \tag{66}$$

$$\lesssim \frac{\epsilon r^{5/2} M}{1-\gamma} E_p\left[\|x-m\|^2\right]^{1/2} \tag{67}$$

$$\lesssim \frac{\epsilon r^{5/2} M}{1-\gamma} E_p\left[\|x-x^*\|^2 + \|x^*-m\|^2\right]^{1/2} \tag{68}$$

$$\lesssim \frac{\epsilon^{3/2} r^{5/2} n^{1/2} M}{(1-\gamma)^2} \tag{69}$$

By the fact that $\|xy^T\| \le \|x\| \cdot \|y\|$ and Jensen's inequality,

$$\left\|\frac{1}{\epsilon}E_p\left[(x-m)R(x)^T + R(x)(x-m)^T\right]\right\| \lesssim \frac{1}{\epsilon}E_p\left[\|x-m\| \cdot \|R(x)\|\right] \tag{70}$$

$$\lesssim \frac{\epsilon^{1/2} r^{5/2} n^{1/2} M}{(1-\gamma)^2} \tag{71}$$

Now, if we choose $\epsilon$ small enough to guarantee the above bound is strictly less than 1, then the Lyapunov equation given in Equation (64) has a unique solution. Furthermore, the Lyapunov equation has an integral form (Särkkä & Solin, 2019), which states that the unique positive definite matrix $\omega$ that satisfies $L\omega + \omega L^T + QQ^T = 0$ can be written as

$$\omega = \int_0^\infty e^{Lt} QQ^T e^{L^T t} dt \tag{72}$$

It's easy to then conclude from the integral form of the Lyapunov equation in Equation (64) that,

$$\|\Sigma/\epsilon - \omega\| \lesssim \frac{\epsilon^{1/2} r^{5/2} n^{1/2} M}{(1-\gamma)^2} \int_0^\infty \|e^{Lt}\| \|e^{L^T t}\| dt. \tag{73}$$

By assumption, $L$ is Hurwitz, and in fact $\|L\| < 1 - \gamma$. By the fact that $\|e^L\| \le e^{\|L\|}$,

$$\|\Sigma/\epsilon - \omega\| \lesssim \frac{\epsilon^{1/2} r^{5/2} n^{1/2} M}{(1-\gamma)^2} \int_0^\infty e^{2\|L\|t} dt \tag{74}$$

$$\lesssim \frac{\epsilon^{1/2} r^{5/2} n^{1/2} M}{(1-\gamma)^2} \int_0^\infty e^{2(1-\gamma)t} dt \tag{75}$$

$$\lesssim \frac{\epsilon^{1/2} r^{5/2} n^{1/2} M}{(1-\gamma)^3} \tag{76}$$

$\square$

PROOF OF THEOREM 4.8

We begin with a series of lemmas. We first need a result about the equilibrium points of the drift, ruling out any degeneracies. As before, we let $x_i^*$ denote the unique zero of $v(x) + c_i = A\sigma(Bx) - x + c_i$.

**Lemma A.5.** *For $r$ independently sampled interventions, $x_i^* - c_i$ are a.s. linearly independent for $1 \leq i \leq r$.*

*Proof.* Let $g(y, z) = \sigma(BAy + z)$. By the contractive constraint on the activation, and the constraints that $\|A\|, \|B\| \leq 1$, it's clear $g$ is a uniform contraction mapping. The derivatives of $g$ satisfy,

$$J_y g(y, z) = J\sigma(BAy + z)BA \tag{77}$$
$$J_z g(y, z) = J\sigma(BAy + z) \tag{78}$$

with all spectral norms bounded by $\gamma < 1$.

By the uniform contraction mapping principle, the fixed point map $y^*(z)$ such that $g(y^*(z), z) = y^*(z)$ is differentiable, with Jacobian

$$Jy^*(z) = (I - J_y g(y^*(z), z))^{-1} J_z g(y^*(z), z) \tag{79}$$

Notice this Jacobian never vanishes, and furthermore $\|y^*(z)\| \to \infty$ as $\|z\| \to \infty$. This follows from the fact that $\sigma' \geq \tau$,

$$\|y^*(z)\| = \|\sigma(BAy^*(z) + z)\| \tag{80}$$
$$\geq \|\sigma(0) + \tau(BAy^*(z) + z)\| \tag{81}$$

which is clearly impossible for $\|z\|$ sufficiently large and $\|y^*\|$ bounded.

Therefore, by the Hadamard global inverse theorem, $y^*(z)$ is a diffeomorphism on $\mathbb{R}^r$.

Now, condition on $A$ and $B$, which are a.s. full rank. Consider the random variable $y^*(Bc)$ where $c$ is a sampled intervention vector. Because $Bc$ has a density in $\mathbb{R}^r$, so does $y^*(Bc)$.

The terms $y^*(Bc_i)$ are sampled independently and hence are a.s. linearly independent. By uniqueness of fixed points, it must be the case that $x_i^* - c_i = Ay^*(Bc_i)$. Therefore, given $r$ interventions, the vectors $x_i^* - c_i$ for $1 \leq i \leq r$ are also a.s. linearly independent.

$\square$

**Lemma A.6.** *The set of vectors $\sigma'(Bx_0^*)/\sigma'(Bx_i^*)$, where $/$ denotes elementwise division, for $1 \leq i \leq r$ are a.s. a basis for $\mathbb{R}^r$. Furthermore, the one-dimensional subspace of of $\alpha \in \mathbb{R}^{r+1}$ such that $\sum_{i=0}^r \alpha_i [\sigma'(Bx_0^*)/\sigma'(Bx_i^*)] = 0$ almost surely satisfies the constraint $\alpha^T \mathbf{1} \neq 0$ for non-zero $\alpha$.*

*Proof.* Again we first condition on $A$ and $B$. By the same argument as in Lemma A.5, we have that $\sigma(Bx_i^*) = y^*(Bc_i)$, and therefore since $\sigma$ is smooth and strictly monotonic, we have that each $Bx_i^* = \sigma^{-1}(y^*(Bc_i))$ has a density and is independent of the other interventions.

Now, additionally conditioning on the value of $\sigma'(Bx_0^*)$, we want to show that $\sigma'(Bx_0^*)/\sigma'(Bx_i^*)$ for each index $i$ are linearly independent.

Define $f(z) = \sigma'(Bx_0^*)/\sigma'(z)$. It's quick to see that $Jf(z)$ is diagonal for any $z$, and rank deficient only when there's some index $j$ such that $\sigma_j''(z)$ is zero. By assumption on $\sigma$, this set of points is measure zero. Hence the Jacobian of $f$ is a.s. full-rank.

By the change of variable theorem induced by the area theorem (Evans, 2018) we have that $f(Bc_i)$ has a density, and therefore the set of vectors $\{f(Bc_i)\}_{i=1}^r$ is a.s. linearly independent.

Finally, let $U$ be the matrix with the columns $\sigma'(Bx_0^*)/\sigma'(Bx_i^*)$ for $1 \leq i \leq r$. The second statement of the lemma now concerns the existence of $\alpha = [\alpha_0, \alpha_1, \ldots \alpha_r]$ such that

$$\alpha_0 \mathbf{1} + U \begin{bmatrix} \alpha_1 \\ \vdots \\ \alpha_r \end{bmatrix} = 0 \tag{82}$$

By above, $U$ is invertible so there is a one-dimensional subspace of satisfying $\alpha$. Then the condition $\alpha^T \mathbf{1} = 0$ is equivalent to $\mathbf{1}^T U^{-1} \mathbf{1} = 1$.

Note this matrix inverse condition can be equivalently written as $\mathbf{1}^T \mathrm{adj}(U)\mathbf{1} = \det(U)$ where $\mathrm{adj}(U)$ denotes the adjugate matrix. This is a non-trivial polynomial equation on a random variable $M$ with density, and therefore it is satisfied only on a set of measure zero, hence a.s. $\alpha^T \mathbf{1} \neq 0$.

$\square$

**Lemma A.7.** *Suppose $X \in \mathbb{R}^{n \times r}$ has full rank and $Y \in \mathbb{R}^{r \times r}$ is invertible. Furthermore, suppose the matrices $D_1, \ldots, D_r \in \mathbb{R}^{r \times r}$ are diagonal, with the vectors along each diagonal linearly independent.*

*Then only observing $X D_i Y$ for $1 \leq i \leq r$, $X$ and $Y$ are uniquely identifiable up to permutation and scaling of their columns and rows, respectively.*

*Proof.* First, note that by the independence assumption, there's some choice of $\alpha \in \mathbb{R}^r$ such that $\sum_{i=1}^r \alpha_i D_i$ has only one non-zero element along its diagonal, and therefore $\sum_{i=1}^r \alpha_i X D_i Y$ is rank-one.

Conversely, suppose $\sum_{i=1}^r \alpha_i X D_i Y$ were rank-one but $\sum_{i=1}^r \alpha_i D_i$ had at least two non-zero elements on the diagonal, w.l.o.g. the first and second elements. We will index $X$ by its columns and $Y$ by its rows. By the independence assumptions, there are vectors $u_1, u_2$ such that $y_i \perp u_1$ iff $i \neq 1$, and likewise for $u_2$. Thus,

$$\left( \sum_{i=1}^r \alpha_i X D_i Y \right) u_1 = X \left( \sum_i \alpha_i D_i \right) e_1 \langle y_1, u_1 \rangle = \beta_1 x_1 \tag{83}$$

$$\left( \sum_{i=1}^r \alpha_i X D_i Y \right) u_2 = X \left( \sum_i \alpha_i D_i \right) e_2 \langle y_2, u_2 \rangle = \beta_2 x_2 \tag{84}$$

where by assumption $\beta_1, \beta_2 \neq 0$. Again by independence, $x_1$ and $x_2$ are not collinear contradicting the rank-one assumption.

Thus, a linear combination of terms that is rank-one offers a certificate that the given linear combination $\alpha$ does in fact yield $\sum_{i=1}^r \alpha_i D_i$ with only one non-zero term. By considering all possible $\alpha$ vectors, one will arrive at each possible rank-one term $\gamma_i x_i y_i^T$ for some unknown scalar $\gamma_i$. $\square$

For the proof of the main theorem, because we have a total of $r + 1$ interventions we will zero-index the interventions and consider $0 \leq i \leq r$ for convenience. We introduce the notation $\sigma'_{[i]}$ to denote the Jacobian of $\sigma$ evaluated at $B x_i^*$, to distinguish it from an element of $\sigma$. Because $\sigma$ acts elementwise, $\sigma'_{[i]}$ is necessarily a diagonal matrix.

*Proof of Theorem 4.8.* By Theorem 4.7, for each intervention in the limit of zero noise we recover the mean vector $x_i^*$ and the covariance matrix $\omega_i$, which satisfies $Jv(x_i^*)\omega_i + \omega_i(Jv(x_i^*))^T + I = 0$. The Jacobian of the vector field satisfies $Jv(x_i^*) = A\sigma'_{[i]}B - I$.

From Lemma A.5, any $r$ terms $x_i^* - c_i$ are linearly independent. Since they are all in the image of $A$, we can derive $P_A$ and hence $P_A^\perp$. From a similar calculation as in the linear case, one can confirm that,

$$2\omega_i P_A^\perp = (I - \tfrac{1}{2}A\sigma'_{[i]}B)^{-1}P_A^\perp, \tag{85}$$

and taking the transpose,

$$2P_A^\perp \omega_i = P_A^\perp (I - \tfrac{1}{2}B^T\sigma'_{[i]}A^T)^{-1}. \tag{86}$$

This matrix is rank $n - r$, so its kernel is dimension $r$ and clearly generated by the columns of

$$(I - \tfrac{1}{2}B^T\sigma'_{[i]}A^T)A \tag{87}$$

Thus, taking an arbitrary basis of the above mentioned kernel must be of the form,

$$Z_i := (I - \tfrac{1}{2}B^T\sigma'_{[i]}A^T)AQ_i, \tag{88}$$

for some invertible matrix $Q_i \in \mathbb{R}^{r \times r}$. Note that here, $Z_i$ is observed but $Q_i$ is not.

Note that,

$$P_A^\perp Z_i = -\frac{1}{2} P_A^\perp B^T \sigma'_{[i]} A^T A Q_i. \tag{89}$$

Observe two facts: 1) $\sigma'_{[i]} A^T A Q_i$ is a.s. an invertible matrix in $\mathbb{R}^{r \times r}$, and 2) $P_A^\perp B^T$ maps from $\mathbb{R}^r$ to an $n - r$ subspace, so if $n > 2r$ then a.s. this is full-rank.

Because $P_A^\perp Z_i$ and $P_A^\perp Z_0$ are a.s. full-rank with the same range, we can derive the unique $M_i$ such that,

$$P_A^\perp Z_i M_i = P_A^\perp Z_0. \tag{90}$$

Furthermore the pseudoinverse $(P_A^\perp B^T)^\dagger$ acts as a left inverse, so we derive the identity,

$$\sigma'_{[i]} A^T A Q_i M_i = \sigma'_{[0]} A^T A Q_0. \tag{91}$$

Notice that $Z_i$ and $M_i$ are observed variables while the $Q_i$ are not. Nevertheless, we can rewrite,

$$Z_i M_i = A Q_i M_i \tag{92}$$

$$= A(A^T A)^{-1} \left( A^T A Q_i M_i \right) \tag{93}$$

$$= (A^\dagger)^T \sigma'_{[0]} (\sigma'_{[i]})^{-1} A^T A Q_0 \tag{94}$$

By Lemma A.6, the diagonal parts of $\sigma'_{[0]} (\sigma'_{[i]})^{-1}$ are linearly independent for $1 \le i \le r$.

Now, by Lemma A.7, we observe the rows of $(A^\dagger)^T$ and the columns of $A^T A Q_0$ up to permutation and scaling. W.l.o.g. we can assume the permuted rows of $(A^\dagger)^T$ are placed in the true order, and we can conclude we observe $(A^\dagger)^T \Lambda_1^{-1}$ and $\Lambda_2 A^T A Q_0$ for some unobserved, diagonal scaling matrices $\Lambda_1$ and $\Lambda_2$.

Taking transpose and another pseudoinverse, we recover $A\Lambda_1$, so it remains to recover $B$.

We can observe,

$$(-2 P_A^\perp Z_0)(\Lambda_2 A^T A Q_0)^{-1} = P_A^\perp B^T \sigma'_{[0]} A^T A Q_0 (\Lambda_2 A^T A Q_0)^{-1} \tag{95}$$

$$= P_A^\perp B^T \sigma'_{[0]} \Lambda_2^{-1}. \tag{96}$$

Taking a transpose, we've observed $\Lambda_2^{-1} \sigma'_{[0]} B P_A^\perp$. Now, returning to equation 85, we can rewrite the observed matrix with Woodbury,

$$2\omega_i P_A^\perp = (I - \frac{1}{2} A \sigma'_{[i]} B)^{-1} P_A^\perp \tag{97}$$

$$= \left( I + \frac{1}{2} A \left( I - \frac{1}{2} \sigma'_{[i]} B A \right)^{-1} \sigma'_{[i]} B \right) P_A^\perp \tag{98}$$

Subtracting $P_A^\perp$, applying the left inverse of $A\Lambda_1$ and the right inverse of $\Lambda_2^{-1} \sigma'_{[0]} B P_A^\perp$ and finally multiplying by two, we observe,

$$\Lambda_1^{-1} (I - \frac{1}{2} \sigma'_{[i]} B A)^{-1} \sigma'_{[i]} (\sigma'_{[0]})^{-1} \Lambda_2 \tag{99}$$

The whole matrix is $\mathbb{R}^{r \times r}$ and invertible, so taking an inverse we calculate,

$$\Lambda_2^{-1} \sigma'_{[0]} (\sigma'_{[i]})^{-1} \Lambda_1 - \frac{1}{2} \Lambda_2^{-1} \sigma'_{[0]} B A \Lambda_1 \tag{100}$$

Now, we claim we can calculate a non-zero $\alpha \in \mathbb{R}^{r+1}$ such that $\sum_{i=0}^r \alpha_i Z_i M_i = 0$. Indeed, by the decomposition in 92, we know this matrix sum is zero if and only if

$$\sum_{i=0}^r \alpha_i \sigma'_{[0]} (\sigma'_{[i]})^{-1} = 0 \tag{101}$$

By lemma A.6, one can find non-zero $\alpha$ satisfying this constraint, and furthermore a.s. $\alpha^T \mathbf{1} \neq 0$. Thus, we can determine $\alpha$ by finding a non-trivial solution to $\sum_{i=0}^{r} \alpha_i Z_i M_i = 0$, because $Z_i$ and $M_i$ are observed.

Consequently, summing equation 100 across several choices for $i$ we can calculate the term:

$$\sum_{i=0}^{r} \alpha_i \left( \Lambda_2^{-1} \sigma'_{[0]} (\sigma'_{[i]})^{-1} \Lambda_1 - \frac{1}{2} \Lambda_2^{-1} \sigma'_{[0]} BA\Lambda_1 \right) = -\frac{\alpha^T \mathbf{1}}{2} \Lambda_2^{-1} \sigma'_{[0]} BA\Lambda_1 \tag{102}$$

Notice that

$$((A\Lambda_1)^T A\Lambda_1)^{-1} (A\Lambda_1)^T = \Lambda_1^{-1} (A^T A)^{-1} A^T, \tag{103}$$

and so multiplying this term on the right and dividing out the non-zero scalar term $-\frac{\alpha^T \mathbf{1}}{2}$, we observe

$$\Lambda_2^{-1} \sigma'_{[0]} BP_A \tag{104}$$

Because we observe $\Lambda_2^{-1} \sigma'_{[0]} BP_A^{\perp}$ as above, and $P_A + P_A^{\perp} = I$, we at last observe $\Lambda_2^{-1} \sigma'_{[0]} B$. Hence we've also recovered $B$ up to scaling of its rows.

$\square$

## B  EXPERIMENTAL DETAILS

### B.1  LINEAR SDE RECOVERY

For the true simulated linear SDE, $A$ and $B$ are sampled with iid Gaussian entries, and then normalized to have spectral norm equal to $0.9$. The true decay matrix $D$ has each diagonal entry sampled iid from the uniform distribution on the interval $[1, 2]$. The model is trained with the loss given in Equation (8), with the Adam optimizer (Kingma, 2014) with an initial learning rate of $0.005$ and $3000$ iterations. Additionally, due to the non-convexity of the objective, on each training instance we run 100 independent initializations and pick the one with the smallest training error. This excludes the oversampled setting with $k = r * \log(n)$, where we get good performance with only 5 independent initializations.

The plotted mean and standard deviations in Figure 2 are based on 5 separate instantiations of the true model and fitting a new model on the stationary distribution. Experiments were run on CPU. All experiments (including those below) were run on a Linux system.

### B.2  NONLINEAR SDE RECOVERY

The true simulated $SDE$ has $A$ and $B^T$ sampled uniformly from the Stiefel manifold, and rescaled by $\gamma = 0.995$, with $\epsilon = 10^{-5}$ and $\sigma$ a sigmoid. We use ambient dimension $n = 8$ and true low-rank dimension $r = 2$. To get training data, we sample $1000000$ samples from each perturbed SDE, initialized at the stationary point with thinning of 300 and burnin of 100. We then calculate the mean and covariance for use in the loss.

In training, the learned network has a MLP to parameterize the learned activation with a hidden dimension of 20, trained for 10000 iterations and learning rate $0.002$ also in Adam, using the loss $\mathcal{L}_{nonlin}$ introduced in Section 5.2. The interventions are iid Gaussians with standard deviation of $0.5$. We do 100 runs, choose the drift with smallest training loss, and measure the normalized Frobenius error against the true matrices $A$ and $B$ (after unit scaling and minimized over all column / row permutations, as the identifiability is only up to scaling and permutation), then report the mean and standard deviation across 20 independent instances.

### B.3  SYNTHETIC NONLINEAR SDE GENERALIZATION

To parameterize learnable activations, given an input $x \in \mathbb{R}^n$, we learn a function that acts elementwise on $x$ without necessarily being the same function on each element, i.e. $\sigma(x) =$

$[\sigma_1(x_1), \ldots, \sigma_n(x_n)]$. We choose to parameterize each $\sigma_i$ as two-layer MLP using the actual sigmoid $\tilde{\sigma}$ as the activation function. As a warm start, we also add $0.1\tilde{\sigma}(x)$ as the initialized activation may be too unstable to train on its own.

We consider a fixed true SDE, with hidden dimension $r = 3$, where the rectangular matrices $A$ and $B^T$ have one down the diagonal and zero elsewhere, $D = I$, $\gamma = 0.98$, and the elementwise activation function is given by

$$\sigma_1(x) = 3\cos(3(x - 0.5))$$
$$\sigma_2(x) = 2\sin(2(x + 1.5)) - 1$$
$$\sigma_3(x) = \sigma_1(x)$$

The hidden dimension of the activation MLPs is fixed at 20.

Each intervention is sampled as a Gaussian vector with mean zero and variance 0.1 on each entry, and from each intervened SDE we collect 5000 samples. To do this, we use the Euler-Maruyama scheme to solve the SDE, and use MCMC to draw approximately independent samples from the stationary distribution, using $dt = 0.01$, a thinning factor of 300 and the first 500 samples treated as burn-in. We use an radial basis function kernel to parameterize the KDS loss.

We train with 10 interventions, and evaluate on 20 held-out interventions. Training is done with AdamW (Loshchilov, 2017), with initial learning rate 0.003 and 50000 iterations. The hyperparameter ranges considered are given in Table 3. All experiments were run on CPU.

Table 3: Hyperparameter ranges considered for synthetic nonlinear SDE experiments.

| Hyperparameter | Range |
|---|---|
| model hidden size $r$ | $\{4, 8, 16\}$ |
| kernel bandwidth | $\{3, 5, 7\}$ |
| $L_1$ weight regularization | $\{0, 10^{-5}, 10^{-4}\}$ |

### B.4 SIMULATED GRN SDE GENERALIZATION

We first give a summary of the simulator. BoolODE is scRNA-seq data simulator that samples mRNA readouts via a stochastic differential equation (SDE). The underlying cyclic GRN is encoded in the SDE via a Hill function based approximation to a boolean logical circuit. $x_i$, $p_i$, and $R[i]$ each represents the mRNA concentration for gene $i$, the concentration of the protein encoded by gene $i$, and the set of proteins that regulate gene $i$. Further, $r$, $l_p$, $l_x$, and $s$ are scalar coefficients representing translation rate, RNA decay rate, protein decay rate, and noise standard deviation respectively. The functions $f_i(\cdot)$ encodes the gene regulatory network. The default SDE is given by,

$$\frac{dx_i}{dt} = f_i(p_{R[i]}) - l_x x_i + s\sqrt{x_i}dB_t \tag{105}$$

$$\frac{dp_i}{dt} = rx_i - l_p p_i + s\sqrt{p_i}dB'_t \tag{106}$$

where $B$ and $B'$ are independent Brownian motions.

We simulate overexpression (e.g., CRISPR-a) experiments by inducing perfect intervention coupled with increased transcription for a set of intervened genes $I_k$. For each intervened gene $j \in I_k$ we set $f_j(\cdot) = 0$ and add a positive shift (set to 20 in the experiments) to $dx_j/dt$. We consider $K$ such intervention regimes, plus the observational setting ($k = 0$) with no intervention, i.e, $I_0 = \emptyset$, for a total of $K + 1$ regimes. With this setup, we obtain a family of distributions $\{\rho_k\}_{k=0}^K$ in gene expression space.

We adapt the neural ODE architecture proposed in Lin et al. (2025). The base SDE model under $I_k$ parametrizes the change of gene expression via the drift function as below. $A$ and $B$ are the coefficient matrices encoding the modular graph. $\alpha$ is a scaling vector controlling the rate of nonlinear activation in the module, where the activation $\sigma$ is the logistic sigmoid. Further, $\beta$ is a bias term that shifts the activation threshold of the modules. The GRN is extracted as $A\,\text{diag}(\alpha \circ \mathbf{1}_N)B$. The intervention is modeled as a combination of standard basis vectors, $\sum_{j \in I_k} e_j$, specifying the

overexpression. The model also learns a single diffusion coefficient scalar. Lastly, $M$ is a masking matrix blocking the signals from the intervened genes' regulators. Altogether, the drift is given by,

$$v_k(x) = M_k A\sigma(\alpha \circ (Bx - \beta)) + \sum_{j \in I_k} e_j - Dx. \tag{107}$$

We alternatively consider the architecture using a learnable MLP to model the nonlinear activation of modular signals,

$$v_k(x) = M_{I_k} A\sigma_*(Bx - \beta) + \sum_{j \in I_k} e_j - Dx \tag{108}$$

where $\sigma_*$ denotes the MLP as described above. The model loss is the Sinkhorn divergence (Feydy et al., 2019) applied to the true perturbed distribution and the samples drawn from the learned SDE.

Each interventional distribution $\rho_k$ is obtained by taking the observational data $\rho_0$ as initial distribution and simulating the SDE via the Euler-Maruyama method.

We fit the hyperparameters by testing the ODE architecture in Lin et al. (2025) on the same simulated datasets, and then doing zero-shot transfer of the hyperparameters to the SDE models. For Bicycle, we sweep the hyperparameters in Table 4 and use the package defaults for the remaining values. Experiments are run on an nvidia tesla V100 gpu.

Table 4: Hyperparameter ranges for GRN simulation.

| Hyperparameter | Range |
|---|---|
| scale $L_1$ | $\{0.0001, 0.001, 0.01, 0.1, 1.0\}$ |
| scale spectral | $\{0, 1.0\}$ |
| scale Lyapunov | $\{0.1, 1, 10\}$ |