# OpenReview forum: "Towards Identifiability of Interventional Stochastic Differential Equations"
_ICLR.cc/2026/Conference — Submitted to ICLR 2026_

### Official Review · Reviewer_229n · 2025-10-17

**Soundness:** 3
**Presentation:** 3
**Contribution:** 1
**Rating:** 2
**Confidence:** 2

**Summary:**

The paper studies identifiability of stochastic differential equations from stationary distributions under multiple interventions. It gives the first theoretical bounds on the number of interventions required in both linear and nonlinear (small-noise) regimes, supported by synthetic and semi-synthetic experiments.

**Strengths:**

•	Rigorous mathematical development with clear proofs.
	•	Theoretical results are novel within the causal inference/SDE literature.
	•	Experiments confirm the identifiability thresholds.

**Weaknesses:**

•	The setting (identifiability from stationary SDEs) is quite narrow and primarily of mathematical interest.
	•	The nonlinear result holds only under restrictive assumptions (contractive drift, small noise).
	•	No real connection is made to learning algorithms or generative diffusion models, which would be essential for ICLR relevance.
	•	The applications is minimal and does not add conceptual depth.

**Questions:**

•	How do the identifiability results for stationary interventional SDEs inform or improve learning algorithms used in practice?
	•	Can the presented theory be applied to diffusion-based generative models, where SDEs define learned data distributions?

---

> ### Author Response · Authors · 2025-11-19
>
> Thank you for your review and your positive notes on the novelty and experimental validity.  We address your further questions and concerns below:
>
> We strongly disagree with the reviewer that the setting is without application, as the task of GRN inference is an extremely important one in genomics [1], and the underlying dynamical model we study has been applied to this task without theoretical guarantees [2,3].  Furthermore, we believe this topic is of interest to ICLR readers.  We give to our knowledge the first result on identfiability for nonlinear SDEs that doesn't assume full information like the generator.  The connection to learning is through the theoretical justification to use learnable activations.  It's not clear a priori which architectures will lead to non-identifiable solutions, and our theory and experiments justify the use of learnable activations in these settings for learning a GRN.
>
> We want to stress that this setup is not related to diffusion models, since our interest is in recovering the dynamics of a physical system given its stationary distribution.  Diffusion models don't meet these criteria as 1) diffusion SDEs typically have either a standard Gaussian or no stationary distribution (variance preserving or variance exploding respectively), and 2) there is no strong incentive to learn the trajectories that diffusion models follow from the noise distribution to the data distribution.  In contrast, understanding the dynamics that cells follow is of great biological interest.
>
> [1] Badia-i-Mompel, Pau, et al. "Gene regulatory network inference in the era of single-cell multi-omics." Nature Reviews Genetics 24.11 (2023): 739-754.
> [2] Lorch, Lars, Andreas Krause, and Bernhard Schölkopf. "Causal modeling with stationary diffusions." International Conference on Artificial Intelligence and Statistics. PMLR, 2024.
> [3] Tong, Alexander Y., et al. "Simulation-Free Schrödinger Bridges via Score and Flow Matching." International Conference on Artificial Intelligence and Statistics. PMLR, 2024.

---

### Official Review · Reviewer_tcP2 · 2025-10-29

**Soundness:** 2
**Presentation:** 2
**Contribution:** 3
**Rating:** 4
**Confidence:** 3

**Summary:**

The paper tackles identifiability of SDE models using only stationary snapshots under multiple interventions (no trajectories). It provides theoretical guarantees for when drift parameters are recoverable from moment information across interventions, covering both linear and nonlinear settings. It validates the ideas on synthetic and semi-synthetic GRN benchmarks.

**Strengths:**

1. **Well-motivated problem.**
The paper addresses the challenge of recovering system dynamics from stationary, intervention-only data, a realistic and important setting for many scientific domains (e.g., biology), where collecting time-series trajectories is often infeasible.
2. **Novel theoretical contribution.**
The work provides, to the best of my knowledge, the first provable identifiability guarantees for SDEs observed only through stationary interventional distributions, covering both linear and nonlinear cases.

**Weaknesses:**

1.  **Theoretical presentation lacks clarity.**
The main theorems are difficult to follow because they do not explicitly list all required assumptions. For instance, Theorem 4.4 depends on distributional/genericity assumptions (Assumption 4.2) and a known $D$ (Assumption 4.3), yet these are not stated in the theorem itself but scattered in the text. This weakens the precision and reproducibility of the claims.
2.  **Restrictive linear setup.**
The linear identifiability results rely on strong and somewhat unrealistic assumptions (e.g., requiring certain structural components or known parameters) which limit their practical applicability and make the “identifiable with $r$ interventions” message feel narrower than presented.
3. **Theory–practice gap in the nonlinear setting.**
The nonlinear guarantees assume globally contractive and monotone activations, but the experiments with learnable activations (generic MLPs) do not appear to enforce these constraints. This creates a noticeable mismatch between the theoretical results and the empirical demonstrations.
4.  **Limited regime of validity.**
The main results hold only in the small-noise regime. The paper itself notes that the proposed losses (e.g., KSD/Sinkhorn) become numerically unstable as noise increases, and that empirical benefits diminish in that regime, which limits the broader applicability.

**Questions:**

1. **Clarify theorem statements.**
Please restate Theorems 4.4 and 4.8 with their assumptions explicitly enumerated (e.g., “Under Assumptions 4.2–4.3 …”; “Under Assumption 4.5, $||A\|, ||B|| \leq 1$, and i.i.d. interventions ..."). This would significantly improve readability and rigor.
2. **Activation constraints in experiments.**
In the nonlinear experiments with learnable activations, were monotonicity or contractivity constraints enforced (e.g., via constrained layers or regularization)? If not, how should readers reconcile the theoretical assumptions with the empirical results?
3. **Sample complexity and scalability.**
Could you report the approximate sample complexity (e.g., cells per intervention) needed for stable recovery in both linear and nonlinear regimes, and discuss how computational cost scales with $n$ and $r$?
4. **Related work suggestion.**
In the “Dynamical System Methods” section of related work, please consider including  _Wang et al., NeurIPS 2023: "Generator identification for linear SDEs with additive and multiplicative noise"_, which provides identifiability results for linear SDEs based on the generator. This paper is highly relevant and would help position your contribution more clearly within the broader identifiability literature.

---

> ### Author Response · Authors · 2025-11-19
>
> Thank you very much for your review and your positive feedback on the work's novelty.  To your weaknesses and questions raised:
>
> 1. We use the assumptions in the order they are mentioned in the paper.  Nevertheless, we are happy to include specific mention of the assumptions made in the theorem for the sake of clarity.
> 2. We disagree that the assumptions in the linear case are unrealistic.  In the linear case, the primary assumptions we are making are: low-rank, which we give considerable evidence for in the genomic setting in the response to reviewer T1DS; known interventions, which in the case of soft interventions is typically necessary for any identifiability guarantees, see for example [1]; and known decay matrix, which can be estimated independently in practice as it corresponds to physical decay of RNA, see [2].
> 3. We disagree this constitutes a substantial disagreement of the theory and practice, as the learnable MLP without constraint is a larger class of functions that still shows approximatey recovery of the underlying parameters.  As for the reason we make this choice, in our experiments we found that enforcing monotonicity and a constrained Lipschitz constant decayed the accuracy of recovering the true parameters in all settings, which we understand as a consequence of the highly non-convex optimization.  Therefore we argue this doesn't discount the identifiability result, but points to the fact that enforcing these constraints in practice creates a bad learning landscape.
> 4. The theory does assume small-noise, but our experiments in section 5.3 demonstrate that the theory still approximately applies when the noise is of non-negligible scale.  On the losses, we'd like to clarify that we do not suffer from instability in practice.  The KDS loss is not unstable when we use it to train, but it can become quite large when testing on unseen interventions (and this is not unique to our setting, as this loss was introduced in [3] where they also use MSE for evaluation).  The Sinkhorn loss is similar.  Crucially, we do not use extremely small noise scales but instead \epsilon \geq 0.05 when using these loss functions, the mention of instability is simply an explanation of why we choose not to use \epsilon = 10^{-5} in section 5.3.
>
> To the questions:
> 1. As above we are happy to make this change, thank you for the suggestion.
> 2. See the answer to point 3 above.
> 3. The identifiability results in the linear case are done in the population setting so there is no sample complexity to assess, and in the non-linear case we use drastic values of \epsilon=10^{-5} and #samples = 10^6 to demonstrate identifiability.  We note that concentration bounds on the empirical covariance (see Thereom 5.6.1 in [4]) suggest the #samples must scale roughly as \epsilon^2, and therefore a large number of samples is necessary when studying the small-noise regime to confirm our theory in ideal conditions.  For more general conditions, we introduce the experiments in section 5.3 on generalization to different interventions as a practical experiment, and here we use only 5000 samples per intervention.  The computational complexity is O(mnr) for all experiments when m is the number of samples, as we have matrices of shape (n, r) in all architectures.
> 4. Thank you for bringing up this work, it is certainly a related setting for SDE identification and we will add discussion in our related work.
>
> [1] Buchholz, Simon, et al. "Learning linear causal representations from interventions under general nonlinear mixing." Advances in Neural Information Processing Systems 36 (2023): 45419-45462.
> [2] Imamachi, Naoto, et al. "BRIC-seq: a genome-wide approach for determining RNA stability in mammalian cells." Methods 67.1 (2014): 55-63.
> [3] Lorch, Lars, Andreas Krause, and Bernhard Schölkopf. "Causal modeling with stationary diffusions." International Conference on Artificial Intelligence and Statistics. PMLR, 2024.
> [4] Vershynin, Roman. "High-dimensional probability." (2009).

---

### Official Review · Reviewer_T1DS · 2025-11-02

**Soundness:** 3
**Presentation:** 3
**Contribution:** 2
**Rating:** 4
**Confidence:** 4

**Summary:**

This paper considers a system with state space R^d and evolution described by a first order stochastic differential equation (SDE), with identity diffusion. The authors assume to have access to 'interventions' which means that the drift coefficient can be modified by addition of a constant term. They ask the question of identifiability of the drift from the stationary distribution of the SDE for multiple values of the intervention.
Two type of results are presented:
1) Linear case. In this case the drift is linear with corresponding matrix given by the sum of a rank r term and an arbitrary known term.
Under a a probabilistic model for the low rank component (that in particular ensures genericity) tyhey prove that r-2 interventions are necessary and r are sufficient.
2) Nonlinear case. The drift is assumed to be parametrized by a two-layer neural network with r hidden neurons. This case is treated in the limit in which the stochastic component of the SDE vanishes, reducing to the linear case by a perturbative argument.

**Strengths:**

In many systems that are modeled by an SDE is unrealistic to assume that we can observe trajectories, and it is instead more common to have access to the stationary distribution. The stationary measure does not uniquely identify the drift and hence the plan of studying this problem under interventions is well motivated and interesting.
The presentation is clear, and the results are easy to understand.

**Weaknesses:**

Establishing identifiability is only the first step towards understanding estimation accuracy; optimal procedures; computational complexity and so on.
The identifiability result in the linear model appears a relatively direct fact of linear algebra.
As for the nonlinear case, the result is purely perturbative and non-quantitative. It requires the drift to be a contraction with a unique fzero, enabling perturbative argument. No quantitative estimate is given on how small \epsilon must be for the identifiability to hold.

**Questions:**

1) Assumption 4.2 is stated in a form that is not very transparent. I believe that what is required is really certain deterministic conditions on A, B, C. Instead the authors choose A,B,C to be random so that those conditions are satisfied a.s. I think it would be much better to express the conditions in deterministic form. Also "each column is drawn iid" probably means that the columns c_1, c_2,.. are iid.

2) Why the low-rank model, and the corresponding r hidden neurons network are good models for the problems proposed in the introduction?

---

> ### Author Response · Authors · 2025-11-19
>
> Thank you for your review!  We address your questions and concerns below:
>
> We agree that identifiability is only one of the desiderata for understanding a parametric method.  However, we want to emphasize that there are no previous analyses of this form for linear or nonlinear SDEs that don't assume knowledge of much stronger properties of the dynamics (i.e. the generator [1] or previously known sparsity in the drift matrix [2]).  Furthermore, it is common in causal inference literature to first establish identifiability without quantitative bounds, as many of the underlying problems are NP-hard and offer few approximation guarantees (see for example [3,4]).
>
> 1. We agree that the distributional assumptions are given as a way to enforce generic conditions almost surely, and the proof could flow the same if we stated deterministic conditions on A, B, C, and the activation \sigma.  However, the proofs require many such conditions that are slightly cumbersome to define.  For example, the linear proof requires a linear dependence between a matrix induced by the solution to Lyapunov's equation and the interventions C, and Lemmas A.5 and A.6 both include conditions on the solutions to fixed point equations induced by the model parameters.  These conditions appear as strong and arbitrary constraints if stated deterministically, even though they hold generically, and we believe it is more transparent to a reader to state a weak condition on the distribution of parameters that guarantees these constraints almost surely.
>
> And yes, as defined on line 115, C is the matrix of concatenated intervention vectors, so assuming each column is iid is equivalently stated as c_i iid for each i.
>
> 2. The motive for using a low-rank intermediate space to parameterize the dynamics comes mainly from the rich literature on genomics.  Many methods inferring gene dynamics, and in genomics in general, first apply PCA and learn dynamics in the projected space [5], which is equivalent to our low-rank assumption.  Alternatively, there are many methods that apply other factorization methods like NMF to learn low-dimensional embeddings under the assumption that genomic data can be explained by few factors.  The biological justification for this step comes from the knowledge of so-called "gene modules", which are sets of genes that act similarly or have correlated expression [6].
>
> Additionally, other works that specifically target cell trajectory inference, model dynamical systems with low-rank parameterizations similar to our setting [7,8].
>
> [1] Hansen, Niels, and Alexander Sokol. "Causal interpretation of stochastic differential equations." (2014): 1-24.
> [2] Dettling, Philipp, et al. "Identifiability in continuous Lyapunov models." SIAM Journal on Matrix Analysis and Applications 44.4 (2023): 1799-1821.
> [3] Squires, Chandler, et al. "Linear causal disentanglement via interventions." International conference on machine learning. PMLR, 2023.
> [4] Buchholz, Simon, et al. "Learning linear causal representations from interventions under general nonlinear mixing." Advances in Neural Information Processing Systems 36 (2023): 45419-45462.
> [5] Luecken, Malte D., and Fabian J. Theis. "Current best practices in single‐cell RNA‐seq analysis: a tutorial." Molecular systems biology 15.6 (2019): e8746.
> [6] Yang, Zi, and George Michailidis. "A non-negative matrix factorization method for detecting modules in heterogeneous omics multi-modal data." Bioinformatics 32.1 (2016): 1-8.
> [7] Lorch, Lars, Andreas Krause, and Bernhard Schölkopf. "Causal modeling with stationary diffusions." International Conference on Artificial Intelligence and Statistics. PMLR, 2024.
> [8] Hossain, Intekhab, et al. "Biologically informed NeuralODEs for genome-wide regulatory dynamics." Genome Biology 25.1 (2024): 127.

---

### Meta-Review · Area_Chair_EFUy · 2026-01-08

**Summary:**

This paper studies when you can identify SDE drift functions using only stationary distributions under constant additive interventions (no time-series/trajectories). It provides:
- Linear case: a low-rank + known-component characterization of how many interventions are needed—roughly necessary around r-2 and sufficient at r under generic conditions.
- Nonlinear case: a result for two-layer neural network drifts, obtained by a small-noise perturbation that reduces the problem to the linear setting.
- Experiments: synthetic tests and semi-synthetic gene regulatory network (GRN) experiments.

**Reviewer Concerns:**

Reviewers think the problem is well motivated (especially for biology) and the linear results/proofs are mostly clear. But they view the overall contribution as too narrow for ICLR because:
- It focuses mainly on identifiability, with little on estimation/algorithms/robustness.
- The linear theory depends on fairly restrictive and sometimes unclear assumptions (genericity phrased in distribution terms, plus assuming known interventions and known decay/extra terms).
- The nonlinear theory is very limited (contractive drift, unique fixed point, small-noise) and non-quantitative (no guidance on how small ε must be).
- Experiments use unconstrained MLPs that don’t really match the stated activation/contractivity assumptions.

The rebuttal clarifies motivation (e.g., low-rank structure in genomics, how decay could be estimated) and promises clearer assumptions, but reviewers feel it doesn’t bridge the theory–practice gap or meaningfully strengthen the nonlinear guarantee beyond the small-noise setting.

**Reviewer Scores:**

- Reviewer T1DS: likely stays 4, with a small chance up to 6. The rebuttal helps on motivation/assumptions, but the nonlinear result is still perturbative and non-quantitative (no clear ε threshold).
- Reviewer tcP2: likely stays 4. Clarity improves, but the main concerns (restrictive assumptions, theory–practice mismatch, small-noise limits) are explained rather than fixed.
- Reviewer 229n: most likely to move to 4. The rebuttal addresses relevance and scope, but probably not enough to reach 4+.

---

### Decision · Program_Chairs · 2026-01-26

Reject